# Exploring the Relationship between Running-Related Technology Use and Running-Related Injuries: A Cross-Sectional Study of Recreational and Elite Long-Distance Runners

**DOI:** 10.3390/healthcare12060642

**Published:** 2024-03-13

**Authors:** Kuntal Chowdhary, Zachary Crockett, Jason Chua, Jennifer Soo Hoo

**Affiliations:** 1Department of Rehabilitation Medicine, New York-Presbyterian Rehabilitation Medicine, Weill Cornell Medical College & Columbia University Irving Medical Center, New York, NY 10032, USA; 2Department of Physical Medicine and Rehabilitation, Harvard Medical School, Boston, MA 02129, USA; zcrockett@mgb.org; 3Spaulding Rehabilitation Hospital, Charlestown, MA 02129, USA; 4Division of Biostatistics, Department of Population Health Sciences, Weill Cornell Medical College, New York, NY 10038, USA; jac7024@med.cornell.edu; 5Department of Clinical Rehabilitation Medicine, Weill Cornell Medicine, New York, NY 10038, USA; jes9343@med.cornell.edu

**Keywords:** running injuries, overuse injury, wearable technology, monitoring technology, fitness tracking

## Abstract

In recent years, the surge in sport and exercise participation, particularly in running, has coincided with the widespread adoption of running-related technology, such as fitness trackers. This study investigates the correlation between the use of running-related technology and running-related injuries among recreational and elite long-distance runners. We conducted a quantitative, cross-sectional online survey of 282 adult runners. Data were analyzed using descriptive statistics and a multivariable logistic regression analysis. Participants, with an average age of 37.4 years, reported varied running experience, with 90.07% utilizing running-related technology during their runs to some degree, primarily smartwatches like Garmin and Apple Watch. Running-related technology users showed a higher likelihood of experiencing running-related injuries compared to non-users (OR = 0.31, *p* < 0.001). However, those who utilized the metrics obtained from running-related technology to guide their training decisions did not exhibit a higher risk of injury. This nuanced relationship highlights the importance of considering individual training behaviors and the potential psychological impacts of technology on running practices. The study underscores the need for future research integrating biomechanical and psychosocial factors into running-related technology to enhance injury prevention strategies.

## 1. Introduction

In recent years, there has been an increase in sport and exercise participation in the United States [1]. Specifically, running is consistently in the top five most popular activities, with participation rates ranging from 7.9 to 13.3 percent globally across the six World Health Organization (WHO) regions [2]. The high participation rates may be attributed to the accessibility of running, given its low cost, small learning curve, and minimal equipment or experience required to start [2,3]. Given the social distancing mandates and several fitness facility closures, the COVID-19 pandemic resulted in a rise in running as a form of exercise in both novice and experienced runners [3,4,5].

Parallel to the growth of exercise participation, there has been an explosion in the fitness technology world, with an increasing number of mobile health (mHealth) apps and devices [6,7,8]. In a study of runners in the Half Marathon Eindhoven 2014, out of the 2172 respondents to the Eindhoven Running Survey (with a response rate of 40.0%), 86.2% of runners had utilized at least one monitoring device during their training in the past 12 months [8]. These running-related technologies (RRTs) allow users to effectively record and measure performance indices such as heart rate, running distance, and training volume [9,10]. Running-related technologies play a crucial role in providing support and monitoring for the substantial population of novice runners lacking professional training and coaching available to elite runners to help guide training decisions.

As novice runners take up the sport, many are often faced with subsequent running-related injuries (RRIs) [11]. Prior epidemiological research indicates that the incidence rates of RRI can reach up to 90% within the running population, predominantly affecting the lower extremities [12]. Furthermore, as many as 79% of these injuries have been classified as overuse or recurrent pathologies, primarily associated with training errors, such as abrupt increases in running volume and intensity [13,14,15,16]. Between 5.6% and 14.8% of runners reported experiencing a new RRI in any given two-week period, with an overall prevalence ranging from 29.2% to 43.5% [17]. Studies have shown certain risk factors for developing RRI, such as body mass index (BMI) greater than 30 kg/m^2^, age between 45 and 65 years, non-competitive (recreational) behavior, previous injuries, and high training volume [18,19]. Indeed, given the propensity for recreational runners to make training errors, such as overtraining with excessive running distance and under-recovery with increased frequency, it is not surprising that a two-times-higher incidence rate of injuries was observed in novice runners, defined as those who had commenced running within the last 12 months [11].

As the number of novice and recreational runners rises and RRTs are more widely adopted, a pivotal question emerges: is there a correlation between the use of RRTs and RRIs? Several empirical studies have explored the modified exercise experiences facilitated by fitness apps, predominantly highlighting the positive health outcomes associated with monitoring physical activity through these technologies [20,21]. Both clinicians and runners are increasingly incorporating RRTs to quantify biomechanics and training loads, aiming to potentially decrease the occurrence of RRIs [7]. Conversely, RRTs can potentially exacerbate adverse behaviors by encouraging obsessive tendencies, resulting in heightened anxiety and the development of exercise addiction patterns. Several fitness apps are crafted to leverage peer pressure, such as sharing workout data and providing virtual praise, as well as promoting social comparisons through features like leaderboards and challenges [22,23]. Engaging in these behaviors may result in training errors, such as overtraining and insufficient recovery, making runners more susceptible to RRIs [22,23].

To date, there have been three studies that evaluated the correlation between RRTs and RRIs in recreational runners. Mayne et al. found that out of 192 runners who participated in a 5 km parkrun, 87.4% used RRTs, with GPS watches being the most common device. Those using RRT ran further and more frequently than non-users. However, there was no significant association between RRT use and the incidence of RRIs [24]. Nielsen et al. evaluated over 7000 runners across 87 different countries and found that the rate of RRIs surpassed 50% after reaching 1000 km [25]. Neal et al. found that the only variable that correlated to RRI in their study of 149 recreational runners utilizing GPS watches was acute load by calculated effort [26].

Although RRI has been studied extensively in various running populations, there is a significant gap in the literature on the study of RRT use and its relation to RRI. Additionally, to the best of our knowledge, there are no studies that have compared RRIs between recreational and elite runners utilizing RRTs. Given that recreational runners have a high risk of RRI, and increased training volume is a strong risk factor for the development of RRI, the aim of our study was to evaluate the correlation between use of RRTs and RRIs in recreational and elite runners who run long distances.

## 2. Materials and Methods

### 2.1. Overview and Study Design

This study was approved by the Weill Cornell Medicine Institutional Review Board (protocol code 20-09022716 and approved on 25 February 2021). All participants provided informed consent for participation. This study used a quantitative, cross-sectional, online survey design to evaluate injuries among long-distance recreational and elite runners utilizing RRTs to track running parameters.

A survey was developed comprising of 4 parts: (a) demographics; (b) running habits; (c) RRT use; and (d) RRI history. The survey was developed through objective peer evaluation to minimize bias in question formulation and was assessed for readability. The survey was based on prior validated tools [27]. The questions were reviewed and discussed with each member of the research team to ensure that bias was minimalized. Initial versions of the survey were given under observation to ensure that any common errors were addressed, with necessary changes made to the survey. Injury was assessed by anatomic region as opposed to specific running-related pathology (e.g., Achilles tendonitis), as these self-reported labels have been reported to be unreliable [28].

#### Definitions

A long-distance run was defined as a run distance greater than 10 km in a single interval.Elite runners were defined as members of professional running groups or collegiate Division I athletes participating in track and field events. This definition was based on the level-of-performance context based on peer participation in the sport, with elite runners participating in a high level of competition. A Division I (DI) athlete refers to a student-athlete engaged in a college sports program affiliated with the National Collegiate Athletic Association (NCAA) Division I. Division I represents the pinnacle of college sports competition in the United States, encompassing numerous prestigious universities and athletic programs renowned for their size and prominence.RRI was defined as an injury sustained from running that caused a modification in distance, pace, or frequency, or which caused the participant to stop running for 7 days or 3 consecutive scheduled training sessions [29]. These injuries were self-reported.RRT is defined as an mHealth phone app or electronic device that is physically worn by individuals in order to track, analyze and transmit personal data.

### 2.2. Participants and Recruitment

Between April 2020 and April 2021, we recruited a convenience sample of elite and recreational runners.

Elite runners were recruited through contacting college and professional cross-country and track and field teams, as well as elite running groups.

Recreational runners were recruited using physical and web-based flyers. These flyers were distributed within the two clinic sites at New York Presbyterian Hospital, on social media platforms (Facebook, Reddit, and X, formerly Twitter), and through recreational running groups across the United States.

To capture a large sample of runners in this study, we utilized broad eligibility criteria. The inclusion criteria were as follows: (1) runners aged 18 years and older; and (2) participants running 10 km or greater in a single interval in the past year. Proficiency in English and internet access were necessary prerequisites for participating in the study.

### 2.3. Data Collection

If runners were interested in participating in the study, they were able to access the online survey through a QR code on the flyer or were provided with an invitation to the survey through an email link. Data for the study were gathered and organized utilizing the REDCap electronic data capture tools (Nashville, TN, USA) hosted at Weill Cornell School of Medicine. Participants were provided contact information for members of the research team and were encouraged to reach out if any questions arose while completing the questionnaire. Please find the recruiting flyers in the Appendix A.

### 2.4. Statistical Analysis

Descriptive statistics (including mean, standard deviation, median, interquartile range, frequency, and percent) were calculated for collected demographic and running/training characteristics and stratified by the two study groups (i.e., runners with and without RRT). The Fisher exact test was used to compare the injury prevalence proportion (i.e., primary endpoint) between the two running groups. The Fisher exact test was also used to compare categorical demographic and running/training characteristics between the two groups. Similarly, the two-sample *t*-test or Wilcoxon rank-sum test was used, as appropriate, to compare continuous demographic variables (e.g., age) between the two running groups. Similar analyses were used to compare demographic and running/training characteristics between recreational and elite (college/professional) runners, as well as to compare RRT usage between recreational and elite runners.

Multivariable logistic regression analyses were used to assess the independent effect of RRT status (i.e., use/non-use) on injury status (i.e., binary outcome variable = injury prevalence over past year), after controlling for demographic and running/training characteristics of interest. Collinearity between predictors in the model were evaluated prior to the formulation of the final multivariable model. Adjusted odds ratios and 95% confidence intervals for smart technology status and demographic/running/training variables of interest were estimated from the multivariable model. All *p*-values were two-sided with statistical significance evaluated at the 0.05 alpha level. Ninety-five percent confidence intervals for all parameters of interest were calculated to assess the precision of the obtained estimates.

All analyses were performed in SAS Version 9.4 (SAS Institute, Inc., Cary, NC, USA) and R Version 3.6.0 (R Foundation for Statistical Computing, Vienna, Austria).

## 3. Results

### 3.1. Demographic Characterisitics

A total of 282 runners met the inclusion criteria and were included in the analysis. Five respondents’ data were excluded as the participants were under the age of 18 years old. The average participant age was 37.4 years old. The sex of the participants was 48.2% female, 51.1% male, and 0.7% responding “Other” (Table 1). The ethnicity of the respondents included 77.7% White, 9.6% Asian/Pacific Islander, 5.3% Hispanic, 5.0% Black/African American, and 2.4% Other (Table 1).

### 3.2. Running Behaviors

Participants were surveyed regarding their lifetime running history with 79 (28.01%) noting 0–5 years of experience, 89 (31.6%) noting 6–10 years of experience, 41 (14.5%) with 11–15 years of experience, 18 (6.4%) noting 16–20 years of experience, and 55 (19.5%) noting 21 or greater years of experience. Of these participants, 67 (23.8%) reported running 1–3 races, 52 (18.4%) reported running 4–6 races, 20 (7.1%) reported running 7–9 races, 43 (15.3%) reported running 10 or greater races, and 25.5% reported being recreational runners that did not participate in organized races (Table 2). Twenty-eight (9.8%) reported being elite runners, running at the collegiate or professional level. Seventy-two (25.5%) reported running recreationally without participating in organized competition (Table 3).

Athletes were stratified based on running experience—recreational and elite—and their training behaviors were analyzed. Elite runners were found to be more likely to train more than 3 days per week (RR = 1.76) and train more than 20 miles per week (OR = 16.48) when compared to recreational runners (Table 3).

### 3.3. Running-Related Technology Use

Most of the individuals that chose to complete the questionnaire used RRT during their runs to some degree (Table 4). Novice runners (0–5 years of running experience) were less likely to use RRT compared to experienced runners (>6 years of running experience) (OR = 0.53). However, novice compared to experienced users were not found to have statistically significant differences in their RRT use (Fisher *p*-Value = 0.5).

Twenty (7.1%) runners used RRT “Sometimes” (<50% of runs), 27 (9.6%) used RRT “Most of the time” (50–75% of runs), 207 (73.4%) used RRT on “All runs”, and 28 (9.9%) reported that they never used RRT on their runs (Table 4). Of these participants, 122 (48.0%) used a smartwatch, 36 (14.2%) used a smartphone application, and 95 (37.4%) used both a smartwatch and a smartphone application (Table 4). The breakdown of smartwatches can be found in Figure 1, which demonstrates 128 (45.4%) using Garmin, 76 (27.0%) using Apple Watch, 16 (5.7%) using Fitbit, 5 (1.8%) using Whoop, 2 (0.7%) using Timex, and 1 (0.4%) using Polar. Additional data regarding metrics tracked and used to return after injury can be found in Table 4.

Participants were stratified by RRT use (“Never”/“Sometimes” vs. “Often”/“Always”) and then analyzed for differences across age (*p* = 0.13), sex (*p* = 0.09), and ethnicity (*p* = 0.63), with no statistically significant findings. Participants that reported “Never”/“Sometimes” using RRT on their runs were also noted to be less likely than those who utilize RRT “Often”/“Always” to train more than 3 days per week (OR = 0.47) or to run for more than 20 miles per week (OR = 0.37). Elite runners were compared to recreational runners and were found to have no statistically significant differences in their smart technology usage (Fisher *p*-Value = 0.5, Table 4). Elite runners were found, however, to use Garmin smart watches (OR = 4.12) more frequently and Apple Watches (OR = 0.09) less frequently than recreational runners. In addition, elite runners were found to more often track data pertaining to activity time (OR = 4.56), altitude change (OR = 2.47), and cadence (OR = 2.43), compared to their recreational counterparts.

### 3.4. Running-Related Injuries

Study participants were asked questions regarding their prior history of running-related injuries. In the last 12 months, 103 (37.0%) study participants reported experiencing an RRI and 177 (63.0%) reporting no running-related injury (Table 5). Respondents were surveyed regarding which anatomical area was affected by injury in Table 6. The lower extremity was the most commonly injury limb, with the most common injuries occurring in the knee (35 runners, 12.4%), foot (27 runners, 9.6%) and lower leg (18 runners, 6.4%).

Participants that reported using RRT “Sometimes” and “Never” were compared to participants that reported using RRT “Most of the Time” and “Always”, regarding their reported rate of RRI in the previous 12 months. The “Sometimes”/“Never” group had an odds ratio of 0.31 (*p* < 0.001) (Table 7). Comparing all runners, those who used RRT were more likely to be injured than those who were not using any smart technology (*p* = 0.02). Those who used smart technology to drive their training decisions were not more likely to be injured than those who did not (*p* = 0.08). Comparing novice runners (0–5 years of running experience) and experienced runners (>6 years of running experience), there was no statistically significant difference in injury incidence (*p* = 0.42). Finally, elite runners were compared to recreational runners and were found to have a statistically significant differences in their incidence of RRI (Fisher *p*-Value = 0.3).

## 4. Discussion

The surge in sport and exercise participation, particularly in running, has led to a parallel growth in fitness technology. These technologies are widely utilized by runners, allowing them to monitor various performance metrics. As novice runners join the sport, many are confronted with RRI, which can have greater than 40% incidence rates within the running population [17]. These injuries, often attributed to training errors, highlight the need for effective injury assessment and prevention strategies. With the evolution of technology, RRTs have emerged as metric libraries, motivational guides, and informal coaches for novice runners lacking professional guidance. However, this begs the question: is there a correlation between the utilization of RRTs and RRIs in runners who run long distances (greater than 10 km)? And, additionally, is there a difference between recreational and elite runners using RRT?

This is the first study to investigate the relationship between RRT and RRI in recreational and elite long-distance runners. Participants in this study, with an average age of 37.4, were younger than in comparable investigations [24,25]. Our findings reveal a significant adoption of RRT among runners, with only 9.93% reporting never using it, reflecting a trend of increased utilization observed in recent studies [24,25]. Most runners primarily utilize either a smartwatch (48.0%) or both a smartwatch and smartphone application (37.4%). Garmin (45.4%) and Apple Watch (27.0%) were the most commonly used smart technologies, with elite runners favoring Garmin (75.0%) over Apple Watch (3.6%). A majority of runners (88.2%) reported using RRT to inform their training decisions, indicating its growing utility compared to previous studies that reported approximately 75% of users utilizing RRT for purposes of training optimization or distance recording [10]. In our comparison of recreational and elite runners, we observed that elite runners tend to monitor specific metrics such as activity time (OR = 4.56), altitude change (OR = 2.47), and cadence (OR = 2.43) more frequently than recreational runners.

In our survey, 37.0% of respondents reported experiencing an RRI in the past year, slightly lower than the 40–50% incidence reported in previous studies [14]. The most commonly affected body parts were the knee (12.4%), foot (9.6%), and lower leg (6.4%), consistent with findings from similar studies [30]. Elite runners were less likely to be injured compared to recreational runners (*p* = 0.03), which corresponds with prior data on risk stratification noting recreational running behavior as a risk factor for RRI [18,19]. Additionally, injury prevalence was found to rise with the distance of the event individuals were training for, aligning with the widely accepted principle linking higher training volumes to increased injury risk among runners [18,19].

Compared to prior studies, our investigation did reveal that, when comparing all runners in the study, those who used RRT were more likely to be injured than those who were not using any RRT (*p* = 0.02). Participants who used RRT “Sometimes” or “Never” had a lower odds ratio for RRI compared to those who used RRT “Most of the Time” or “Always” (OR = 0.31, *p* < 0.001).

These findings have crucial implications for runners, whether for a casual recreational runner or devoted elite runner. RRT use can predispose a runner to a higher likelihood of RRI, particularly in the case of recreational and inexperienced runners. Based on metrics provided by RRTs, recreational and inexperienced runners may acutely increase their training loads. In a study by Janssen et al. evaluating how different types of runners utilized RRTs, they found that the highest proportion of runners utilizing RRT metrics for ongoing training monitoring and to make adjustments to their training regimen were individual competitive recreational runners [31]. The individual competitive recreational runners are thus at high risk of injury, as they lack the professional coaching to safely increase training loads and are intrinsically driven to ramp up their running speeds and distances. Furthermore, these negative behaviors can be worsened in this population by fostering obsessive tendencies, leading to increased anxiety and the formation of exercise addiction patterns. Many fitness apps are designed to capitalize on peer pressure, encouraging users to share workout data and receive virtual praise, while also facilitating social comparisons with features like leaderboards and challenges [22,23]. Furthermore, the “carry over” effect of negative emotions, such as “guilt” or “disappointment”, may be heightened, especially among inexperienced users of self-tracking apps, impacting their decisions related to activity tracking [32].

In their qualitative assessment, Janssen et al. also found that the minority of runners who did not use RRTs had varied reasons for abstaining. Their concerns ranged from considering running with a device as “ignorant” to a desire for a more authentic running experience where technology did not play a role [31]. In a survey of the German running community conducted by Wiesner et al., it was discovered that among non-RRT users, over two-thirds mentioned relying on their body’s cues rather than technology as the primary reason for not using RRTs. Some also expressed privacy concerns [33].

Despite the prevalence of RRI, studies indicate that the use of RRT for injury assessment and prevention is an emerging field, poised to advance as technology continues to evolve. Our research has demonstrated a correlation between the use of RRTs and an increased incidence of RRIs. Moreover, we observed a significant portion of our sample (44.5%) utilizing RRTs to inform their training choices. Consequently, there is a pressing need to enhance the design and functionality of RRTs with a specific focus on injury prevention. RRTs that emphasize integrating biomechanics, training frequency, intensity, non-running physical activity, recovery, sleep quality, or psychosocial factors can aid in identifying preventive measures for RRI [34]. Specifically, RRTs that assess movement patterns or training loads linked to injury, identify individuals at risk for future injury or re-injury, monitor training loads and their outcomes, and offer real-time feedback on movement patterns to aid in the rehabilitation of runners with RRIs, should be taken into account [7]. Our study lays the foundation for future studies to enhance RRTs with a user-centered focus of injury prevention. Additionally, further studies are needed to elucidate the precise connection between utilizing RRTs for training decisions and the occurrence of RRIs.

### Limitations

The utilization of the online questionnaire format through REDCap software Version 10.8.3 enabled participants to be surveyed from regions beyond the investigation’s location, facilitating the inclusion of a broader range of social and geographic demographics. However, the absence of investigators during participant survey completion may have resulted in instances where participants unintentionally provided incorrect information. Moreover, the substantial proportion of participants selecting “Other” in the “Event Typically Trained For” field hindered a comprehensive assessment of how training variables might have contributed to injury, as further clarification was not obtained. Responses regarding recent injury may have been subject to recall bias, potentially leading participants who experienced injury to recall other training variables contributing more accurately to their RRI. Lastly, the relatively modest sample size of 282 participants completing the online questionnaire suggests the need for additional studies to delve deeper into the relationship between training variables, RRT, and RRI.

## 5. Conclusions

Our study sheds light on the intersection of RRT utilization and RRI among recreational and elite long-distance runners. Our findings underscore the significant adoption of RRT in the running community, with Garmin and Apple Watch emerging as the most favored devices. Additionally, elite runners were less likely to be injured compared to recreational runners. Notably, while RRT users tended to be more likely to experience RRI compared to non-users, those who employed RRT to inform their training decisions did not exhibit a higher risk of injury. This suggests a nuanced relationship between RRT use and injury susceptibility, particularly in novice and recreational runners. Thus, it is crucial to consider individual training behaviors and the potential psychological impacts of technology on running practices. Future studies integrating biomechanical and psychosocial factors into RRTs could enhance injury prevention strategies and promote safer running practices for all levels of runners. Additionally, further research with larger sample sizes and a longitudinal design is warranted to deepen our understanding of these complex dynamics.

## Figures and Tables

**Figure 1 healthcare-12-00642-f001:**
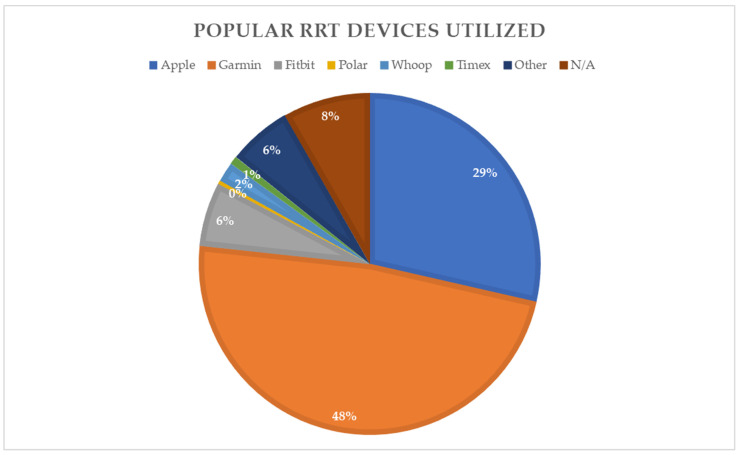
Popular RRT Devices Utilized.

**Table 1 healthcare-12-00642-t001:** Demographics: Elite vs. Recreational Runners.

	Elite Runners (*n* = 28)	Recreational Runners (*n* = 254)	Total (*N* = 282)
**Age (SD)**	23 (9.4)	38.91 (14.4)	37.41 (14.7)
**Sex (%)**			
*Male*	14 (50)	130 (51.2)	144 (51.4)
*Female*	13 (46.4)	123 (48.4)	136 (48.6)
*Other*	1 (3.6)	1 (1.9)	
**Ethnicity (%)**			
*Asian/Pacific*	1 (3.6)	26 (10.2)	27 (9.6)
*Black*	1 (3.6)	13 (5.1)	14 (5.0)
*Hispanic*	1 (3.6)	14 (5.5)	15 (5.3)
*White*	24 (85.7)	195 (76.8)	219 (77.7)
*Other*	1 (3.6)	6 (23.6)	7 (2.5)

**Table 2 healthcare-12-00642-t002:** Running Behaviors: All Runners.

**Running History**	** *N* **	**Percent**
0–5 years	79	28.0
6–10 years	89	31.6
11–15 years	41	14.5
16–20 years	18	6.4
21+ years	55	19.5
**Races per Year**	** *N* **	**Percent**
Run recreationally to stay fit ^1^	72	25.5
Run 1–3 races per year	67	23.8
Run 4–6 races per year	52	18.4
Run 7–9 races per year	20	7.1
Run 10+ races per year	43	15.3
Run at the elite level ^2^	28	9.9

^1^ Do not participate in organized competition; ^2^ Collegiate or professional level.

**Table 3 healthcare-12-00642-t003:** Running Behaviors: Elite vs. Recreational Runners.

	Elite Runners (*n* = 28)	Recreational Runners (*n* = 254)	Total (*N* = 282)
**RRT use (%)**			
*Yes*	26 (92.9)	228 (89.8)	254 (90.1)
*No*	2 (7.1)	26 (10.2)	28 (9.9)
**Events trained for (%)**			
*10 km*	15 (7.5)	134 (52.8)	149 (52.8)
*15–20 km*	0 (0)	33 (13.0)	33 (11.7)
*Half-Marathon*	2 (7.1)	132 (52.0)	134 (47.5)
*25–30 km*	0 (0)	5 (2.0)	5 (1.8)
*Marathon*	1 (3.6)	82 (3.2)	83 (29.4)
*Ultramarathon*	2 (7.1)	39 (15.4)	41 (14.5)
*Ironman*	0 (0)	5 (2.0)	5 (1.8)
**Miles run per week (%)**			
*0–10*	0 (0)	63 (24.8)	63 (22.3)
*11–20*	2 (7.1)	79 (31.1)	81 (28.7)
*21–30*	2 (7.1)	49 (19.3)	51 (18.1)
*31–40*	4 (14.3)	26 (10.2)	30 (10.6)
*>=41*	20 (71.4)	37 (14.6)	57 (20.2)

**Table 4 healthcare-12-00642-t004:** RRT Use: Elite vs. Recreational Runners.

	Elite Runners (*n* = 28)	Recreational Runners (*n* = 254)	Total (*N* = 282)
**Frequency of use (%)**			
*Never (0%)*	2 (7.1)	26 (10.2)	28 (10.0)
*Sometimes (<50%)*	0 (0)	20 (7.9)	20 (7.1)
*Most of the time (50–75%)*	3 (10.7)	24 (9.4)	27 (9.6)
*All the time (100%)*	23 (82.1)	184 (72.4)	207 (73.4)
**RRT Type (%)**			
*SmartWatch*	13 (46.4)	109 (42.9)	122 (48.3)
*SmartApp*	1 (3.6)	35 (13.8)	36 (14.2)
*Both*	11 (39.3)	84 (33.1)	95 (37.4)
*N/A*	1 (3.6)	0 (0)	1 (0.4)
**Metrics Tracked (%)**			
*Activity Time*	26 (93.0)	188 (74.0)	214 (75.9)
*Heart Rate*	16 (57.1)	156 (61.4)	172 (61.0)
*Heart Rate Zone*	8 (28.6)	64 (25.2)	72 (25.5)
*Respiratory Zone*	2 (7.1)	9 (3.5)	11 (3.9)
*Distance*	26 (93.0)	219 (86.2)	245 (86.9)
*Pace*	26 (93.0)	200 (78.7)	226 (80.1)
*Calories*	7 (25.0)	111 (43.7)	118 (41.8)
*Altitude Change*	12 (42.9)	59 (23.2)	71 (25.2)
*Cadence*	12 (42.9)	60 (23.6)	72 (25.5)
*Stride Length*	4 (14.3)	18 (7.1)	22 (7.8)
*Ground Contact*	2 (7.1)	9 (3.5)	11 (3.9)
*Distance Improvement*	2 (7.1)	29 (11.4)	31 (11.0)
*Pace Improvement*	3 (10.7)	50 (19.7)	53 (18.8)
*Other*	1 (3.6)	6 (2.4)	7 (2.5)
**Influence Training Decisions (%)**			
*Yes*	8 (2.9)	105 (41.3)	113 (44.5)
*Sometimes*	17 (61.0)	94 (37.0)	111 (43.7)
*No*	1 (3.6)	29 (11.4)	30 (11.8)
**Believe RRT helped prevent injury**	16 (57.1)	107 (42.1)	123 (43.6)

**Table 5 healthcare-12-00642-t005:** RRI: Elite vs. Recreational Runners.

	Elite Runners (*n* = 28)	Recreational Runners (*n* = 254)	Total (*N* = 282)
**RRI in last 12 months (%)**	15 (53.6)	88 (34.6)	103 (36.9)
**History of prior RRI (%)**	15 (53.6)	85 (33.5)	100 (35.8)

**Table 6 healthcare-12-00642-t006:** Area of RRI.

	Injured	Not Injured
**Lower Back (%)**	11 (3.9)	271 (96.1)
**Hamstring (%)**	10 (3.6)	272 (96.1)
**Knee (%)**	35 (12.4)	247 (87.6)
**Lower Leg (%)**	18 (6.4)	264 (93.6)
**Achilles Tendon (%)**	17 (6.0)	265 (94.0)
**Ankle (%)**	13 (4.6)	269 (95.4)
**Foot (%)**	27 (9.6)	255 (90.4)
**Other (%)**	19 (6.7)	263 (93.3)

**Table 7 healthcare-12-00642-t007:** RRI: All Runners.

	Injured	Non-Injured	*p*-Value
**Age**			
*>30 years*	69	99	**0.02 ***
*≤30 years*	31	83	
**Miles run per week**			
*>20 Miles*	63	75	**0.0005 ***
*≤20 Miles*	37	107	
**RRT use**			
*Yes*	96	158	**0.013 ***
*No*	4	24	
**Sex**			
*Male*	44	92	0.37
*Female*	54	90	
**Use tracked parameters to guide training decisions**			
*Never (0%)*	4	24	**0.03 ***
*Sometimes (<50%)*	4	16	
*Most of the time (50–75%)*	11	16	
*All the time (100%)*	81	126	

* Statistically significant results.

## Data Availability

Dataset available on request from the authors.

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
