# Peer review of "Exploring the Relationship between Running-Related Technology Use and Running-Related Injuries: A Cross-Sectional Study of Recreational and Elite Long-Distance Runners"

_healthcare, 2024, doi:10.3390/healthcare12060642_

Round 1

Reviewer 1 Report

Comments and Suggestions for Authors

Dear respected authors,

I read the article with considerable interest. After careful evaluation, I regret to inform you that your manuscript is not suitable for publication in the Healthcare. The manuscript has the logical goal of providing an overview of running-related technology (RRT) use its correlation with running-related injuries (RRI) in sports. However, the article's quality does not meet the necessary standards for publication in Healthcare. There are three main reasons for my decision.

First, the intro is not clearly presented and it has too many gaps for the RRT tools and their potential effects on the exercise performance. As such, the background lacks a convincing theoretical framework regarding the main topic, which is pretty popular in exercise sciences.

Second, I could not see anything about the validity of the survey, which was used to collect data for RRT and RRI, and therefore I can not consider it a reliable tool for such research.

Lastly, the conclusions do not provide convincing findings for the readers. This may limit the interest and relevance of the paper to the readership.

Author Response

Dear Reviewer,

Thank you for your evaluation of our manuscript. We have carefully revised the manuscript based on other reviewers' feedback, which has significantly improved its quality.

We have addressed the clarity of the introduction, provided validity information on the survey methodology, and enhanced the conclusions to present more convincing findings.

Reviewer 2 Report

Comments and Suggestions for Authors

First of all, congratulations for the work done, then I will mention a number of changes and recommendations in order to obtain clearer and more accurate information.

- Comments on the abstract:

Avoid the use of abbreviation in the abstract.

- Comments on the introduction:

References must precede the dot

Avoid the use of abbreviation at the beginning of a sentence. Line 49.

“Engaging in these behaviors may result in training errors, such as over-training and insufficient recovery, making runners more susceptible to RRIs” What is the reference to do this statement?

Line 81. “Mayne et al” There is a missing dot. Check all these references.

- Comments on material and methods:

“This study was approved by the Weill Cornell Medicine Institutional Review Board” Write the approval number of the ethics committee.

Who assessed the survey?

Lines 116-117. Check grammar, there are some mistakes… “an mHealth”, “electronic devices that is physically 116 worn”…

You do not mention any normality test.

- Comments on results:

Line 179. Table 2 must be before the dot.

Line 181. Table 3 must be before the dot.

Line 191. The same with table 4.

In method you talk about chi-square test, then, in results you talk about Fisher test.

-Comment on conclusion:

Reading the conclusion, it is not clear to me whether it answers the research question posed in the objective.

-General comments:

You should revise all the manuscript, check all references, there are all bad placed after the dot. The English and grammar need a correction throughout the manuscript.

Comments on the Quality of English Language

The English and grammar need a correction throughout the manuscript.

Author Response

Dear Reviewer,

  1. Summary

We would like to thank the reviewer for the constructive comments and feedback on our manuscript. We sincerely appreciate the response and feel that our subsequent changes have increased the quality of this manuscript. We have prepared a point-by-point response to each comment below. Please see our responses after each comment.

  1. Point-by-point response to Comments and Suggestions for Authors

Comment 1: Avoid the use of abbreviation in the abstract.

Response 1: We have removed abbreviations from the abstract.

Comment 2: References must precede the dot

Response 2: This has been revised.

Comment 3: Avoid the use of abbreviation at the beginning of a sentence. Line 49.

Response 3: This has been revised.

Comment 4: “Engaging in these behaviors may result in training errors, such as over-training and insufficient recovery, making runners more susceptible to RRIs” What is the reference to do this statement?

Response 4: References have been added.

Comment 5: “Mayne et al” There is a missing dot. Check all these references.

Response 5: This has been revised.

Comment 6: “This study was approved by the Weill Cornell Medicine Institutional Review Board” Write the approval number of the ethics committee.

Response 6: The protocol code and approval date have been added. These are also found under the Institutional Review Board Statement on page 11.

Comment 7: Who assessed the survey?

Response 7: Clarification on design and assessment of the survey have been added (lines 107-113).

Comment 8: Lines 116-117. Check grammar, there are some mistakes… “an mHealth”, “electronic devices that is physically 116 worn”…

Response 8: Since the letter “M” in mHealth is pronounced starting with a vowel (‘em’), the article “an” is conventionally used to precede it. Please advise if you agree. We can make the necessary changes if you prefer the changes be made. Device has been made singular.

Comment 9: You do not mention any normality test.

Response 9: Normality tests were not completed on the data for a few reasons:

  1. Since the n is small, normality tests have little power to reject the null hypothesis. Consequently, it's common for small samples to yield non-significant results in normality tests.
  2. The only continuous variable to perform normality tests on was age; however, we treated age as a binary variable based on the distribution. Therefore, normality tests were not indicated.
  3. Finally, for these reasons, we utilized the Fischer’s Exact Test, which is used for small sample size and is non-parametric; therefore, it makes no assumptions about the normality of the data.

Comment 10: Line 179. Table 2 must be before the dot.

Line 181. Table 3 must be before the dot.

Line 191. The same with table 4.

Response 10: This has been revised.

Comment 11: In method you talk about chi-square test, then, in results you talk about Fisher test.

Response 11: We have corrected this in the text.

Comment 12: Reading the conclusion, it is not clear to me whether it answers the research question posed in the objective.

Response 12: Our research question sought to explore the relationship between the use of RRTs and RRIs in long-distance recreational and elite runners. We have further refined the conclusion to reflect this objective.

Comment 13: You should revise all the manuscript, check all references, there are all bad placed after the dot. The English and grammar need a correction throughout the manuscript.

Response 13: The manuscript has been further revised to address these issues.

  1. Additional clarifications

To the Academic Editor: While it's reasonable to view the inclusion of brand names as potentially endorsing or diverting attention from the paper's main focus, our objective remains to provide comprehensive epidemiologic data on RRT, encompassing the range of devices utilized. Consequently, delving into specific brands can offer a clearer picture of the technological landscape. If the editor strongly opposes the inclusion of smartwatch brand names, we are open to removing them.

Reviewer 3 Report

Comments and Suggestions for Authors

Thank you for submitting send manuscript “Exploring the Relationship Between Running-Related Technology Use and Running-Related Injuries: A Cross-Sectional Study of Recreational and Elite Long-Distance Runners”. Presenting the research results of the submitted scientific article required many sacrifices and logistics and organization of scientific research. After reviewing mentioned above scientific paper I would like to require many advice and changes.

Below I am sending You outline possible points for revision in the chronological order of the manuscript.

Scientific paper - In whole scientific paper authors should correct the all citations, each of citation [3] should end with a full stop [3]. and not as it is in the whole article . [3]

Keywords - reduce keywords to a maximum of 4-5, authors in mentioned scientific paper add to much keywords (9 words).

Introduction
a) authors are using words (they) it is not scientific language. Please change it for example: line 46 “the participants”.

b) Line 51 - lack of citations.                                                                               

c) Line 67-69 – the authors asked the question “As the number of novice and recreational runners rises and RRTs are more widely 67 adopted, a pivotal question emerges: Is there a correlation between the use of RRTs and 68 RRIs?” If authors wanted to formulate some questions it should be added to the section of: Methods.                                                                                                                    
d) Line 78-79 – the authors added the following sentence “Engaging in these behaviors may result in training errors, such as over- 78 training and insufficient recovery, making runners more susceptible to RRIs” – It could be added to the discussion not to the introduction.                                                           
e) Line 80-81 – lack of citations.

f) Line 95 – the authors should define in the Introduction the meaning of “run long-distance”.          

g) Line 110 – the authors should explain the meaning of “collegiate Division 1”.                              

h) Line 113-115 - the authors should add the information “whether the doctor diagnosed the injury or injuries”.

 i) In the section of “Participants and Recruitment” – the authors should add information about the age of each group.                                                                                                                                               
j) Results. What was the purpose of dividing the professional and amateur groups - there are too many differences in numbers between the groups in Table 1. It is unreasonable to compare results when, for example, in the elite group in the trained tab the difference between the groups is 15 to 134? It is difficult to compare and justify that the results obtained can be compared. A similar situation occurs in Table 3. Meanwhile in the Table 2 the authors made one group of both – it is difficult to access the behavior when in the group we have got the elite and amateur athletes.                                      

k) Line 197-199. The authors in one sentence use word Table 4 twice – please correct it.              

l) Figure 1 and 2 – it will be more readable to present the results of the research in a table divided into the mentioned study groups.                                                                                                                                 
m) Despite the enormous amount of work that the authors have put into writing the discussion, this section should be thoroughly revised. The authors should increase the number of citations in order to discuss the research results obtained, rather than just presenting their research findings.           

n) References – please check the following references: 1, 3, 4, 5, 6, 8, 10, 13, 18, 19, 20, 22, 29, 30, 31, 32 – lack of pages of scientific journals.

Comments on the Quality of English Language

Except for a few stylistic or grammatical errors, the article reads very smoothly.

Author Response

Dear Reviewer,

  1. Summary

We would like to thank the reviewer for the constructive comments and feedback on our manuscript. We sincerely appreciate the response and feel that our subsequent changes have increased the quality of this manuscript. We have prepared a point-by-point response to each comment below. Please see our responses after each comment.

  1. Point-by-point response to Comments and Suggestions for Authors

Comment 1: In whole scientific paper authors should correct the all citations, each of citation [3] should end with a full stop [3]. and not as it is in the whole article . [3]

Response 1: This has been corrected throughout the document.

Comment 2: reduce keywords to a maximum of 4-5, authors in mentioned scientific paper add to much keywords (9 words).

Response 2: We have reduced keywords to 5.

Comment 3: authors are using words (they) it is not scientific language. Please change it for example: line 46 “the participants”.

Response 3: This has been revised.

Comment 4: Line 51 - lack of citations.

Response 4: References have been added.

Comment 5: Line 67-69 – the authors asked the question “As the number of novice and recreational runners rises and RRTs are more widely 67 adopted, a pivotal question emerges: Is there a correlation between the use of RRTs and 68 RRIs?” If authors wanted to formulate some questions it should be added to the section of: Methods.

Response 5: Thank you for the suggestion. The question emerges of whether a correlation exists between RRTs and RRIs. This inquiry aims to involve the reader within the scope of the study's objectives. It's not a direct query posed to the participants, but rather serves to segue into the objectives outlined in the introduction. This stylistic approach was chosen by the authors for audience engagement, but is open to revision or removal.

Comment 6: Line 78-79 – the authors added the following sentence “Engaging in these behaviors may result in training errors, such as over- 78 training and insufficient recovery, making runners more susceptible to RRIs” – It could be added to the discussion not to the introduction.  

Response 6: This has been further explored in the discussion (lines 296-313).

Comment 7: Line 80-81 – lack of citations.

Response 7: Citations have been added.

Comment 8: Line 95 – the authors should define in the Introduction the meaning of “run long-distance”.

Response 8: This information is found under the section entitled “Definition” in lines 116-117.

Comment 9: Line 110 – the authors should explain the meaning of “collegiate Division 1”. 

Response 9: The definition has been provided in lines 121-125.

Comment 10: Line 113-115 - the authors should add the information “whether the doctor diagnosed the injury or injuries”.

Response 10: These were self-reported injuries; we have clarified this in the manuscript.

Comment 11: In the section of “Participants and Recruitment” – the authors should add information about the age of each group.

Response 11: The average age and standard deviation of the runners can be found in Results 3.1 and in Table 1.

Comment 12: What was the purpose of dividing the professional and amateur groups - there are too many differences in numbers between the groups in Table 1. It is unreasonable to compare results when, for example, in the elite group in the trained tab the difference between the groups is 15 to 134? It is difficult to compare and justify that the results obtained can be compared. A similar situation occurs in Table 3. Meanwhile in the Table 2 the authors made one group of both – it is difficult to access the behavior when in the group we have got the elite and amateur athletes.    

Response 12: The tables were divided into elite and recreational runners to evaluate for differences between the two groups, as our objective is to evaluate a correlation between RRT use and RRI between these groups. We had fewer elite runners participating in the study as compared to recreational runners. For this reason, in addition to providing the absolute values, we have also provided percentages. Table 2 provides a broad overview of running behaviors in all runners, whereas Table 3 delves into the running behaviors of all elite vs recreational runners. Table 2 can be moved to the supplementary data if this helps with the readability.

Comment 13: Line 197-199. The authors in one sentence use word Table 4 twice – please correct it.

Response 13: This has been addressed.

Comment 14: Figure 1 and 2 – it will be more readable to present the results of the research in a table divided into the mentioned study groups.  

Response 14: A table has been added to replace Figure 2.

Comment 15: Despite the enormous amount of work that the authors have put into writing the discussion, this section should be thoroughly revised. The authors should increase the number of citations in order to discuss the research results obtained, rather than just presenting their research findings.

Response 15: Our discussion comprises two distinct sections. The initial four paragraphs compare our findings with those of similar studies. Subsequently, the following three paragraphs delve deeper into the implications of our findings, drawing upon psychosocial factors, qualitative data, and references to aid in the development of a more user-centered RRT aimed at injury prevention in runners. We can formalize these divisions within the discussion to enhance readability.

It's worth noting that this field of research is relatively novel, which accounts for the scarcity of literature, as outlined in our introduction. We have conscientiously cited high-quality articles with significant scientific impact. Increasing the number of citations to include lower quality and poorly written manuscripts would compromise the quality of this manuscript. We understand your concerns and are amenable to citing sources for specific statements that you feel are indicated.

Comment 16: References – please check the following references: 1, 3, 4, 5, 6, 8, 10, 13, 18, 19, 20, 22, 29, 30, 31, 32 – lack of pages of scientific journals.

Response 16: Please note, reference 1 is a website, therefore it will lack pages. References 3, 4, 5, 8, 13, 19, and 22 are from electronic sources (i.e. electronic journals) that lack page numbers. Page numbers have been added for references 6, 10, 18, 30 (now 32), 31 (now 33). Page numbers have already been included for reference 20 and 32 (now 34).

  1. Additional clarifications

To the Academic Editor: While it's reasonable to view the inclusion of brand names as potentially endorsing or diverting attention from the paper's main focus, our objective remains to provide comprehensive epidemiologic data on RRT, encompassing the range of devices utilized. Consequently, delving into specific brands can offer a clearer picture of the technological landscape. If the editor strongly opposes the inclusion of smartwatch brand names, we are open to removing them.

Round 2

Reviewer 2 Report

Comments and Suggestions for Authors

I am pleased to see that the suggestions and corrections have been made correctly.

Regarding the suggested correction in "an mHealth", it can be as you put it, I usually read it as mobile health although it is abbreviated, hence the suggested change, but it seems correct what you indicate. 

I only have one suggestion, in this citation format, the citation is better with a space between the previous word and the citation (You have this: "In recent years, there has been an increase in sport and exercise participation in the 35 United States [1]", and it would be more correct like this: "In recent years, there has been an increase in sport and exercise participation in the 35 United States [1]").

Reviewer 3 Report

Comments and Suggestions for Authors

The authors corrected most of the revisions and incorporated the submitted corrections into the scientific article, therefore the scientific publication "Exploring the Relationship Between Running-Related Technology Use and Running-Related Injuries: A Cross-Sectional Study of Recreational and Elite Long-Distance Runners" fulfils the requirements for publication in a scientific journal.